# Community-based prevalence of rheumatic heart disease in rural Ethiopia: Five-year follow-up

Tadesse Gemechu[1], Eldryd H. O. Parry[2], Magdi H. Yacoub[3,4], David I. W. Phillips[5], Susy Kotit[3]*

1 Jimma University Hospital, Jimma, Ethiopia, 2 London School of Hygiene and Tropical Medicine, London, United Kingdom, 3 Aswan Heart Centre, Aswan, Egypt, 4 NHLI, Heart Science Centre, Imperial College London, London, United Kingdom, 5 Developmental Origins of Health and Disease Division, University of Southampton, Southampton General Hospital, Southampton, United Kingdom

* susykotit@hotmail.com

**Data Availability Statement:** Data is available in the Southampton University data repository, https://doi.org/10.5258/SOTON/D1987.

## Abstract

### Background

As little is known about the prevalence and clinical progression of subclinical (latent) rheumatic heart disease (RHD) in sub-Saharan Africa, we report the results of a 5 year follow-up of a community based, echocardiographic study of the disease, originally carried out in a rural area around Jimma, Ethiopia.

### Methods

Individuals with evidence of RHD detected during the baseline study as well as controls and their family members were screened with a short questionnaire together with transthoracic echocardiography.

### Results

Of 56 individuals with RHD (37 definite and 19 borderline) in the original study, 36 (26 definite and 10 borderline) were successfully located 57.3 (range 44.9–70.7) months later. At follow-up two thirds of the definite cases still had definite disease; while a third had regressed. Approximately equal numbers of the borderline cases had progressed and regressed. Features of RHD had appeared in 5 of the 60 controls. There was an increased risk of RHD in the family relatives of borderline and definite cases (3.8 and 4.0 times respectively), notably among siblings. Compliance with penicillin prophylaxis was very poor.

### Conclusions

We show the persistence of echocardiographically demonstrable RHD in a rural sub-Saharan population. Both progression and regression of the disease were found; however, the majority of the individuals who had definite features of RHD had evidence of continuing RHD lesions five years later. There was an increased risk of RHD in the family relatives of borderline and definite cases, notably among siblings. The findings highlight the problems

**Funding:** Funded by a grant to TG, MHY and DIWP by Chain of Hope www.chainofhope.org. The funders had no role in study design, data collection and analysis, decision to publish, or preparation of the manuscript.

**Competing interests:** The authors have declared that no competing interests exist.

faced in addressing the problem of RHD in the rural areas of sub-Saharan Africa. They add to the evidence that community-based interventions for RHD will be required, together with appropriate ways of identifying active disease, achieving adequate penicillin prophylaxis and developing vaccines for primary prevention.

## Author summary

Although chronic rheumatic heart disease is one of the most important causes of heart disease in sub-Saharan Africa it is still poorly understood. In particular, little is known about the frequency, severity and progression of the disease in the rural areas where most of the population live. In 2017, we reported the results of a community-based study in a rural area of south-west Ethiopia using ultrasound imaging of the heart (echocardiography) which showed a high prevalence of the disease. In a follow-up of the cases originally identified in the study, we found that there was considerable variability of the disease with some individuals showing progression and others regression of the heart abnormalities. The majority of affected individuals, however, had evidence of continuing disease five years later. We also found an increased risk in the family relatives of cases, notably among siblings. Although affected individuals were instructed to take prophylactic penicillin to prevent the disease progressing, compliance with this was very poor. The findings highlight the problems faced in addressing rheumatic heart disease in rural Africa. They suggest that community-based interventions will be required together with innovative ways of identifying active cases, achieving adequate antibiotic prophylaxis and primary prevention.

## Introduction

RHD remains a major health care problem, especially in unprivileged populations, with regional differences in prevalence, severity and course of the disease. However, the regional prevalence, potential contributing factors, severity and rate of progression and/or regression are poorly understood, especially in rural areas.[1,2]

 In 2017, we reported the results of a community-based prevalence study among children and young adults living in a rural area of Jimma, south-west Ethiopia [3], which showed a high prevalence of different degrees of RHD in the community. We here present a follow-up study of the original cohort of the affected patients as well as matched controls from the community in order to clarify the course of the disease and analyse the rate of progression and/or regression of subclinical latent valvular disease over time.

## Methods

### Ethics statement

The Jimma University ethical review board approved the project and written, informed consent was obtained from each participant.

### Fieldwork

The original study [3] was carried out in the Jimma Zone of Oromia regional state, south-west Ethiopia and used a multi-stage, cluster-sampling technique to derive a sample of 987

participants, aged 6 to 25 years. The prevalence was 37.5 cases per 1000 increasing to 56.7 cases per 1000 when borderline cases were included.

In the current study, all the cases previously identified as having definite or borderline RHD were contacted together with a group of controls of similar age and gender composition to the cases, which had been screened during the first study and did not have any features of RHD. Controls (1–2 per case) were the next available individuals in the survey matched by gender and age (within two years). All subjects that accepted our invitation for follow-up were revisited. A questionnaire was verbally administered in *Oromifa* (the local language) for inclusion in the registry updating the data previously collected on demographic, social and medical history, together with details of adherence to penicillin prophylaxis. It was followed by a brief clinical examination and echocardiography.

The composition of the households and, specifically, first degree relatives of subjects in the cohort (definite, borderline and a sample of controls) were registered and all first-degree relatives invited to attend the screening clinic. They also completed a brief questionnaire and clinical examination before screening by transthoracic echocardiography.

## Echocardiography

A transthoracic echocardiogram was carried out with a CX50 Phillips portable echo machine to obtain parasternal long-axis views, parasternal short axis, apical four chamber, apical long axis and five chamber views. Abnormalities in valve morphology and the presence of valve regurgitation with Doppler interrogation were classified using the current World Heart Federation criteria to ascertain whether definite or borderline disease was present. [4] Definite RHD was defined as the combination of pathological regurgitation with at least two morphological features of RHD or mitral stenosis with a mean gradient ≥4 mmHg or borderline disease of both the aortic and mitral valve. Morphological features of RHD for the mitral valve were anterior mitral leaflet or chordal thickening, restricted leaflet motion or excessive leaflet tip motion during systole; for the aortic valve the features were irregular or focal thickening, a coaptation defect, restricted leaflet motion or prolapse. Borderline disease was defined as having either (a) at least two morphological features of RHD of the mitral valve without regurgitation or stenosis or (b) pathological mitral or aortic regurgitation. Images from both the case series and controls were reviewed by a second experienced cardiologist.

## Statistical analysis

Data were double entered before analysis with SPSS version 26. Data are presented as cross-tabulations or mean +/- SD values and comparison between groups carried out with a $\chi^2$ test. Relative risks, confidence intervals and p-values for the analysis of the family data were calculated according to the method of Altman. [5]

## Results

The follow-up study was carried out at a median of 57.3 (range 44.9–70.7) months after the original study when the participants had a mean age of 17.0 (SD 3.6) years.

Of the individuals who were originally diagnosed as having definite (n = 37) or borderline (n = 19) disease we were able to locate 26 (70%) and 10 (53%) respectively. One of the participants had died, this was due to non-cardiac causes. Eight had migrated to another area (largely due to marriage). The remainder were unavailable mainly as a result of work commitments. Table 1 shows the progression or regression of echocardiographic features in the participants who were followed up according to their original diagnostic category. Of the 26 cases with definite disease, 9 (35%) had regressed with six subjects no longer showing valvular disease while

**Table 1. Progression or regression of echocardiographic changes in the Ethiopian subjects followed up a mean of 4.7 years after original screening.**

| | Original echocardiogram classification | | |
| --- | --- | --- | --- |
| | **Normal** | **Borderline** | **Definite** |
| No of subjects | 60 | 10 | 26 |
| Gender M/F | 30/30 | 7/3 | 13/13 |
| Age at follow up, yr (SD) | 16.6(3.4) | 15.3(2.5) | 18.3(4.0) |
| Duration of follow-up, yr (SD) | 4.8(0.3) | 4.7(0.6) | 4.6(0.5) |
| Echocardiogram classification at follow-up | | | |
| Normal (%) | 55(92) | 4(40) | 6(23) |
| Borderline (%) | 2(3) | 3(30) | 3(12) |
| Definite (%) | 3(5) | 3(30) | 17(65) |

17 (65%) continued to have definite lesions of RHD. Of the 10 borderline subjects, 4 showed evidence of regression and 3 progressed to have definite features.

Of the 23 cases identified with definite RHD at follow-up, 17 had mitral valve disease, three had aortic valve disease and three mixed valve disease. Three had an audible murmur, 13 described having palpitations although all were in sinus rhythm when examined, and 12 complained of shortness of breath on exertion with two having demonstrable ankle oedema (at the original examination only three of the 23 subjects complained of breathlessness while four had noted palpitations). None of the subjects had required cardiac intervention and no cardiac mortality was reported.

A total of 60 controls were also successfully located over the same follow-up time. Of the controls 5 (8.3%) developed borderline (n = 3) or definite (n = 2) disease (Table 1). The majority of these (4 out of 5) reported episodes of sore throat, but no other features specific for ARF.

Participants who had shown evidence of regression of lesions (n = 13) had a similar age/sex distribution to those whose echocardiographic status was unchanged (n = 20) or showed evidence of progression (n = 8). They also had similar parental educational and occupational status (S1 Table).

Compliance for penicillin prophylaxis was reported in 2 out of 36 individuals (5.6%) with definite or borderline lesions, although all definite and borderline cases were advised to do so. In both cases a parent worked within the health service as a health extension worker.

Of the 23 participants who had definite RHD at follow-up, 20 agreed to have their families screened and of a potential total of 94 first degree relatives 66 (70%) attended for echocardiographic examination. Screening rates were similar for the families of borderline cases and controls. Table 2 shows the numbers of cases detected in households according to proband status and the nature of the relationship of the affected individuals.

Among 66 individuals screened from 20 households where the proband had definite RHD, 13 (20%) had definite disease, a further six (9%) borderline disease while the remaining 47 subjects were normal. In comparison, of 43 subjects from families of normal probands only 2 (5%) had definite RHD and a further one borderline disease ($\chi^2$ = 7.7, p = 0.02). Table 3 shows the relative risks for RHD according to the status of the proband. For probands who had definite RHD, the risk of definite disease in the family relatives was 3.1 times greater and for the combination of borderline and definite disease 2.3 times greater. Where the proband had borderline or definite disease, the risks of RHD in the family relatives were higher (3.8 and 4.0 times respectively). Most of the increased risk was in the siblings; the numbers of cases in children or parents was too small to derive stable risk estimates.

**Table 2. RHD case detection among the first degree relatives according to proband status in 39 households.**

| Status of proband | No of households | Number of first degree relatives examined | Relationship | Mean age (yr) | Echo diagnosis of relatives | | |
|---|---|---|---|---|---|---|---|
| | | | | | Normal (%) | Borderline (%) | Definite (%) |
| Normal | 13 | 43 | All | | 40(93) | 1(2) | 2(5) |
| | | | Sibling | 12.2 | 23 | 1 | 1 |
| | | | Parent | 44.7 | 17 | 0 | 1 |
| | | | Child | - | 0 | 0 | 0 |
| Borderline | 8 | 20 | All | | 15(75) | 3(15) | 2(10) |
| | | | Sibling | 11.8 | 7 | 1 | 0 |
| | | | Parent | 42.6 | 6 | 2 | 2 |
| | | | Child | - | 0 | 0 | 0 |
| Definite | 20 | 66 | All | | 47(71) | 6(9) | 13(20) |
| | | | Sibling | 16.8 | 30 | 4 | 7 |
| | | | Parent | 39.5 | 15 | 1 | 5 |
| | | | Child | 10.3 | 2 | 1 | 1 |

## Discussion

This study serves to define the rate of progression and/or regression of valvular lesions in a rural area of Sub-Saharan Africa previously shown to have particularly high prevalence of RHD [6,7,8]. Valvular lesions persisted in the majority (65%) of children or young adults originally diagnosed as having definite disease. Although none of the subjects had yet required specific cardiac treatment and no cardiac deaths were reported, a significant number of the patients identified with definite disease reported clinical symptoms and may require intervention in the future. The study was based on the screening of a large number of children and young adults who were carefully selected to be representative of the rural population in this area of Oromia regional state in southern Ethiopia. While the study is small, locating patients in remote rural areas is difficult and very time-consuming and our overall tracing rate of 67% after nearly five years is therefore a major achievement.

Our study does raise questions as to the usefulness of active community screening programs especially in remote rural areas. Apart from the major disadvantage of needing expensive equipment and a high degree of training, many patients identified in our study had mild disease with most remaining asymptomatic during the course of the study. Also the survey data were unable to predict which subjects had continuing or progressive disease. In addition, as has been reported widely [9,10] there was considerable heterogeneity of the echocardiographic appearances over the course of the study with evidence of both progression and regression of

**Table 3. Risk of RHD in family relatives according to the status of the proband.**

| Proband status | Relatives evaluated | Risk of RHD in relatives | | | |
|---|---|---|---|---|---|
| | | Definite | | All | |
| | | RR(CI)+ | p-value | RR (CI) | p-value |
| Definite RHD | All family: | 3.1(1.1–9.0) | 0.04 | 2.3(1.1–4.8) | 0.03 |
| | Siblings: | 5.6(0.7–43.5) | 0.09 | 3.0(0.9–9.7) | 0.08 |
| Any RHD* | All family: | 3.8(0.9–15.7) | 0.07 | 4.0(1.3–12.5) | 0.02 |
| | Siblings: | 3.6(0.5–27.4) | 0.20 | 3.1(0.7–12.6) | 0.12 |

*Any RHD: definite or borderline disease in proband.

+Relative risk and 95% confidence interval.

valve lesions and the appearance of new disease in 8% of people who had normal echocardiograms during the initial screening.

Disappointingly, although all the definite and borderline cases were advised to receive regular penicillin, almost all reported that they did not. Poor compliance is a widespread problem, [11] and adequate penicillin prophylaxis compliance is extremely hard to achieve. This is likely due to factors common to similar remote rural regions, including lack of training of nurses and health officers in local health centres, poor patient or family understanding of the necessity of penicillin, poor access and high relative costs of accessing health centres and obtaining antibiotic treatment, supply problems and the discomfort associated with i.m. injections. These findings highlight the problems faced in addressing the problem of RHD globally and the need for better primary prevention through the development of an effective vaccine. [12,13,14]

Early observations that RHD exhibits familial clustering have been amply confirmed although these studies failed to reach a consensus on the nature of the underlying susceptibility.[15] Undoubtedly, the very strong links with socioeconomic disadvantage and risk factors shared by members of the same household are likely to contribute towards household clustering. Although our dataset is small we show that in 20 households where the proband had definite RHD, almost 30% of the first degree relatives screened also had borderline or definite RHD. This was much greater than the control group where only 7% of screened relatives had evidence of disease. Our results (Table 3) may be compared with a study in Uganda, which showed that children with any RHD were 4.5 times as likely to have a sibling with definite RHD. [16] The increased risk of RHD in family members does however encourage the possibility of screening first-degree relatives once an affected person has been identified especially in endemic areas where health care resources are limited.

In summary, we show the persistence of echocardiographically demonstrable RHD in a rural sub-Saharan population. Although we found evidence of both regression and progression of the disease, the majority of individuals who had definite features of RHD had evidence of continuing RHD lesions. This study emphasises the fact that there are still many unknown contributing factors for the development and progression of RHD valvular lesions, and that efforts should be made to better understand the mechanisms related to all types of susceptibility and develop effective primary prevention in the form of vaccines.

## Supporting information

**S1 Table. Age/Sex and parental education/occupation in subjects whose cardiac lesions regressed, were unchanged or progressed during the five years of the study.**
(DOCX)

## Acknowledgments

The authors are particularly grateful for the encouragement and support from Lisa Yacoub.

## Author Contributions

**Conceptualization:** Tadesse Gemechu, Eldryd H. O. Parry, Magdi H. Yacoub, David I. W. Phillips.

**Data curation:** David I. W. Phillips, Susy Kotit.

**Formal analysis:** David I. W. Phillips, Susy Kotit.

**Funding acquisition:** Magdi H. Yacoub.

**Investigation:** Tadesse Gemechu.

**Methodology:** Eldryd H. O. Parry, Magdi H. Yacoub, David I. W. Phillips, Susy Kotit.

**Project administration:** Tadesse Gemechu.

**Supervision:** Eldryd H. O. Parry, Magdi H. Yacoub.

**Validation:** Susy Kotit.

**Writing – original draft:** David I. W. Phillips, Susy Kotit.

**Writing – review & editing:** Magdi H. Yacoub, David I. W. Phillips, Susy Kotit.

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
