## [Decision Letter · Decision Letter 0]

3 Aug 2021

Dear Dr Kotit,

Thank you very much for submitting your manuscript "Community-based prevalence of Rheumatic Heart Disease in rural Ethiopia: five-year follow-up" for consideration at PLOS Neglected Tropical Diseases. As with all papers reviewed by the journal, your manuscript was reviewed by members of the editorial board and by several independent reviewers. In light of the reviews (below this email), we would like to invite the resubmission of a significantly-revised version that takes into account the reviewers' comments. 

Thank you for the opportunity to consider your manuscript, "Community-based prevalence of Rheumatic Heart Disease in rural Ethiopia: five-year follow-up." The outcome of screen-detected RHD is indeed of importance, and natural history studies, though inherently flawed in many ways, contribute meaningfully to our growing pool of evidence around this condition. Below you will see reviewer comments from 3 experts in this field, who raise important issues that would strengthen the presentation of these data. Please consider these comments when preparing your revision.

Sincerely,

Dr. Andrea Beaton

We cannot make any decision about publication until we have seen the revised manuscript and your response to the reviewers' comments. Your revised manuscript is also likely to be sent to reviewers for further evaluation.

Sincerely,

Liesl Joanna Zuhlke

Deputy Editor

Liesl Zuhlke

Deputy Editor

Thank you for the opportunity to consider your manuscript, "Community-based prevalence of Rheumatic Heart Disease in rural Ethiopia: five-year follow-up." The outcome of screen-detected RHD is indeed of importance, and natural history studies, though inherently flawed in many ways, contribute meaningfully to our growing pool of evidence around this condition. Below you will see reviewer comments from 3 experts in this field, who raise important issues that would strengthen the presentation of these data. Please consider these comments when preparing your revision.

Sincerely,

Dr. Andrea Beaton

Reviewer's Responses to Questions

**Key Review Criteria Required for Acceptance?**

**Methods**

-Are the objectives of the study clearly articulated with a clear testable hypothesis stated?

-Is the study design appropriate to address the stated objectives?

-Is the population clearly described and appropriate for the hypothesis being tested?

-Is the sample size sufficient to ensure adequate power to address the hypothesis being tested?

-Were correct statistical analysis used to support conclusions?

-Are there concerns about ethical or regulatory requirements being met?

Reviewer #1: No

Reviewer #2: Yes. Objectives, study design are well described. Sample volume was not applicable since it is a follow up study of RHD patients.

Reviewer #3: The authors present 5-year follow-up data on patients with subclinical RHD from Ethiopia. Theycompare progression and regression rates with a similar number of controls. It would be useful for the authors to provide details on how they chose the controls. Were they chosen randomly? What was the sampling population? Was any matching performed? What was the rationale for chosing the number of subjects that they did?

**Results**

-Does the analysis presented match the analysis plan?

-Are the results clearly and completely presented?

-Are the figures (Tables, Images) of sufficient quality for clarity?

Reviewer #1: No

Reviewer #2: Results are described in detail. Authors should also mention the number of cases that were clinically diagnosed based on auscultation initially.

Reviewer #3: The reported estimates of progression and regression are consistent with the published literature. For the subjects who showed progression, readers may wish to know more details regarding how the disease progressed. What were the lesions? How many valves were affected? Were the lesions moderate or severe?

**Conclusions**

-Are the conclusions supported by the data presented?

-Are the limitations of analysis clearly described?

-Do the authors discuss how these data can be helpful to advance our understanding of the topic under study?

-Is public health relevance addressed?

Reviewer #1: No

Reviewer #2: Conclusions are supported by the data. Limitations described but not in detail. The loss of follow up could affect the results of the study, since the numbers are significant.

Reviewer #3: The greater frequency of subclinical RHD among siblings may reflect the effect of common exposures rather than genetic influences. Therefore, the authors may wish to place less emphasis on the reasons for this finding, and instead discuss the practical implications for screening programs in greater detail.

**Editorial and Data Presentation Modifications?**

Reviewer #1: (No Response)

Reviewer #2: Minor modifications are mentioned in above sections.

Reviewer #3: (No Response)

**Summary and General Comments**

Reviewer #1: In this study, the authors describe a study that followed up children diagnosed with latent RHD in 2017.

After 57 months of follow up, two thirds of 36 patients who they were initially identified as definite still remained in the definite category, while a third regressed and some had started developing symptoms of clinical disease.

Comments

While it is impressive that the authors were able to track the study participants five years after the initial screening, there are several issues to address:

What was the rationale for the follow up? Was it a natural history study? What was the sample size estimation and did it account for the changes in valve morphology at study end? Were these participants followed up between year 1 and year 5, were there any strategies for surveillance for worsening including recurrences of acute rheumatic fever? How was equipoise regarding BPG administration to latent RHD participants handled?

This study reflects a general approach to latent RHD screening that was done 2010 and 2015. Typically, cross sectional screening for latent RHD identified children with various grades of latent RHD according to the 2012 WHF and were then followed up. Most of the earlier similar studies have been cited by the authors.

The earlier studies, including this one, had several limitations including lack of follow up to detect episodes of acute rheumatic fever recurrence that could have led to progression of the valve lesion, lack of standardised administration of benzathine penicillin, no proper sample size estimation to allow for advanced statistics to objectively measure the stated outcomes such as progression and regression of valve lesions. Consequently, these designs have been over taken by more recent studies with better planned sample size and follow up plan that allow for proper estimation of outcomes including changes in valve lesion morphology. Secondly, until we have evidence on the role of BPG on latent RHD outcomes, blinded screening and follow up may not be useful as this will generate information already known from the natural history of RF/RHD.

Reviewer #2: It is one of several papers available on follow up of subclinical or latent RHD. However authors have in addition studied the family members of both RHD cases and controls.

Reviewer #3: (No Response)

PLOS authors have the option to publish the peer review history of their article (what does this mean?). If published, this will include your full peer review and any attached files.

Reviewer #1: No

Reviewer #2: Yes: Anita Saxena

Reviewer #3: No
---

## [Decision Letter · Decision Letter 1]

21 Sep 2021

Dear Dr Kotit,

We are pleased to inform you that your manuscript 'Community-based prevalence of Rheumatic Heart Disease in rural Ethiopia: five-year follow-up' has been provisionally accepted for publication in PLOS Neglected Tropical Diseases.

Best regards,

Andrea Beaton, MD

Guest Editor

Liesl Zuhlke

Deputy Editor

The authors have addressed most of the major critiques from round 1 of reviews. The manuscript has been sufficiency strengthened with appropriate acknowledgment of areas of limitations and with reasonable conclusions based on these limitations. We now find this manuscript acceptable for publication. I will point out that there are remaining concerns around data availability that should be addressed - please see the PLOS policy: "The PLOS Data policy requires authors to make all data underlying the findings described in their manuscript fully available without restriction, except in cases where the data are legally or ethically restricted (for example, participant privacy is an appropriate restriction)."

---

## [Editor Report · Acceptance letter]

30 Sep 2021

Dear Dr Kotit,

We are delighted to inform you that your manuscript, "Community-based prevalence of Rheumatic Heart Disease in rural Ethiopia: five-year follow-up," has been formally accepted for publication in PLOS Neglected Tropical Diseases.

Best regards,

Shaden Kamhawi

co-Editor-in-Chief

Paul Brindley

co-Editor-in-Chief
